# Affine-Equivariant Kernel Space Encoding for NeRF Editing

**Mikołaj Zieliński** [1]  **Krzysztof Byrski** [2]  **Tomasz Szczepanik** [2]  **Dominik Belter** [1]  **Przemysław Spurek** [2 3]

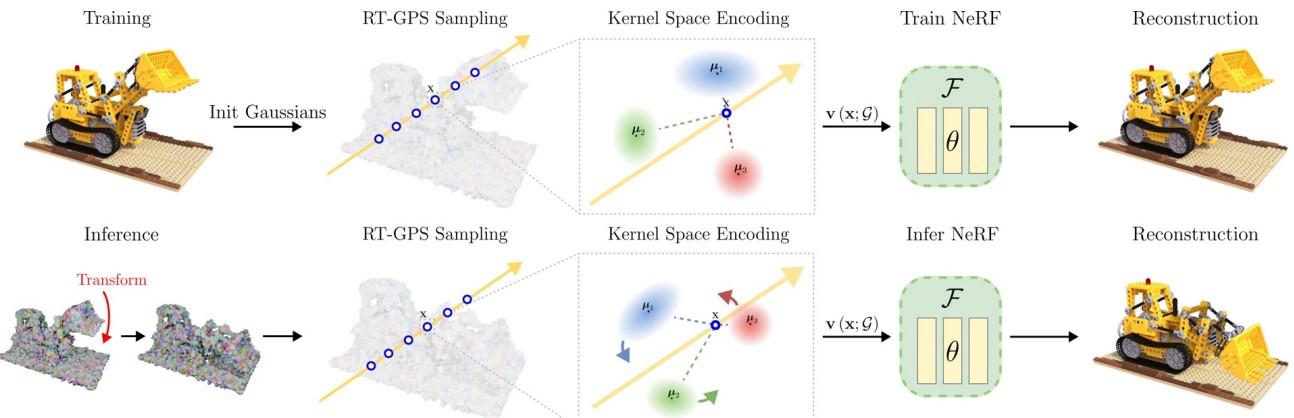

*Figure 1.* **EKS overview.** EKS represents positional features using spatially localized anisotropic Gaussian kernels, enabling stable and fine-grained interactive editing while maintaining the high-fidelity rendering of Neural Radiance Fields.

## Abstract

Neural scene representations achieve high-fidelity rendering by encoding 3D scenes as continuous functions, but their latent spaces are typically implicit and globally entangled, making localized editing and physically grounded manipulation difficult. While several works introduce explicit control structures or point-based latent representations to improve editability, these approaches often suffer from limited locality, sensitivity to deformations, or visual artifacts. In this paper, we introduce Affine-Equivariant Kernel Space Encoding (EKS), a spatial encoding for neural radiance fields that provides localized, deformation-aware feature representations. Instead of querying latent features directly at discrete points or grid vertices, our encoding aggregates features through a field of anisotropic Gaussian kernels, each defining a localized region of influence. This kernel-based formulation enables stable feature interpolation under spatial transformations while preserving continuity and high reconstruction quality. To preserve detail without sacrificing editability, we further propose a training-time feature distillation mechanism that transfers information from multi-resolution hash grid encodings into the kernel field, yielding a compact and fully grid-free representation at inference. This enables intuitive, localized scene editing directly via Gaussian kernels without retraining, while maintaining high-quality rendering. Code can be found under (https://github.com/MikolajZielinski/eks).

## 1. Introduction

Recent years have seen rapid progress in 3D scene representation and rendering, driven by applications in robotics, virtual environments, and content creation that increasingly demand physically grounded simulation and interactive editing (Wang et al., 2023a; Authors, 2024; Huang et al., 2024). Tasks such as object manipulation, deformable modelling, collision handling, and physics-aware animation require 3D representations that are both high-fidelity and intuitively editable while remaining compatible with physics engines.

Neural Radiance Fields (NeRFs) (Mildenhall et al., 2020)

[1]Poznan University of Technology, Institute of Robotics and Machine Intelligence, ul. Piotrowo 3A, Poznań 60-965, Poland [2]Jagiellonian University, Faculty of Mathematics and Computer Science, Łojasiewicza 6, 30-348, Krakow, Poland [3]IDEAS Research Institute. Correspondence to: Mikołaj Zieliński <mikolaj.zielinski@put.poznan.pl>.

*Proceedings of the 43$^{rd}$ International Conference on Machine Learning*, Seoul, South Korea. PMLR 306, 2026. Copyright 2026 by the author(s).

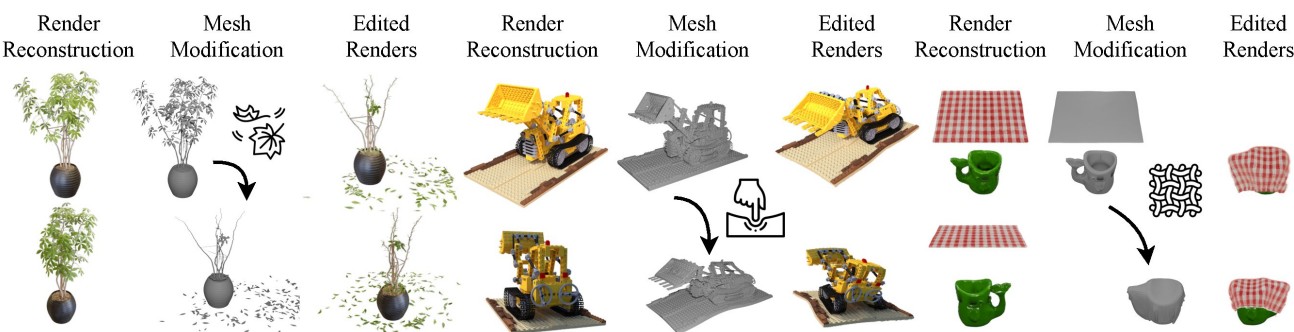

*Figure 2.* **Physical simulations.** From left to right: (1) Rigid body simulation of falling leaves. (2) Soft body simulation of the Lego dozer being squished. (3) Cloth simulation of fabric falling onto a cup. The middle columns show the deformation-driving meshes.

achieve high visual fidelity by modelling scenes as continuous volumetric functions capable of high-quality novel-view synthesis and complex view-dependent effects. However, NeRFs encode spatial structure implicitly within network parameters, making localized scene edits difficult to perform without retraining (Wang et al., 2023a; Weber et al., 2024). This limitation restricts their applicability in interactive and physically grounded settings. Several works seek to mitigate this issue by introducing explicit control structures, including point-based conditioning (Wang et al., 2023b; Chen et al., 2023; Zhang et al., 2023), mesh-based control (Yuan et al., 2022; Yang et al., 2022), or primitive-based representations for simulation (Monnier et al., 2023). While these approaches enable limited forms of manual editing, they are typically constrained to coarse modifications and often introduce visual artifacts. Recent advances in explicitly parametrized scene representations demonstrate that spatial locality and explicit structure can substantially improve editability and interaction (Kerbl et al., 2023; Malarz et al., 2025; Borycki et al., 2024). These results highlight desirable properties for editable 3D representations, but do not directly address how such locality can be integrated into NeRF models. Motivated by these observations, we address a fundamental limitation in NeRF editing task: the absence of a transformation-aware space encoding. Existing NeRF encodings, including positional encodings (Mildenhall et al., 2020) and multi-resolution hash grids (Müller et al., 2022), entangle features globally across space, causing localized modifications to propagate undesirably and preventing precise control. Prior attempts to alleviate this issue rely on point-based latent representations that store features at discrete spatial locations and interpolate them at query points (Xu et al., 2022; Chen et al., 2023; Wang et al., 2023b). While this enables explicit spatial manipulation, such representations remain highly sensitive to deformations, as changes in point positions alter local neighbourhood and lead to unstable interpolation (see Fig. 4).

In this work, we introduce Affine-**E**quivariant **K**ernel **S**pace Encoding (EKS), a novel positional encoding mechanism for NeRFs. Our approach represents scene features in a continuous kernel space, where each latent element defines a spatially localized and anisotropic region of influence. This formulation replaces discrete point samples and grid-based embeddings with a variant kernel field that supports stable and localized feature evaluation. In practice, kernels are parametrized as anisotropic Gaussians, enabling efficient feature evaluation via a $k$-nearest neighbour search weighted by the Mahalanobis distance. By accounting for local anisotropy through kernel covariances, this interpolation remains stable under spatial transformations while capturing richer local geometric structure than point-based representations. To retain the expressiveness of multi-resolution encodings, we further propose a training-time feature distillation mechanism that transfers spatial detail from hash grid encodings into the kernel field. Unlike prior approaches that embed Gaussian features within fixed grids (Govindarajan et al., 2024), the resulting representation is fully decoupled from grid structures at inference time, yielding an editable, deformable, and compact latent field that preserves high reconstruction quality.

## 2. Related Works

Several approaches focus on modeling deformation or displacement fields at a per-frame level (Park et al., 2021a;b; Tretschk et al., 2021; Weng et al., 2022), while others aim to capture continuous motion over time by learning time-dependent 3D flow fields (Du et al., 2021; Gao et al., 2021; Guo et al., 2023; Cao & Johnson, 2023). While these methods effectively model dynamic scenes, they primarily focus on temporal consistency rather than controllable geometry editing.

A substantial body of research has also explored NeRF-based scene editing across different application domains. This includes methods driven by semantic segmentation or labels (Bao et al., 2023; Dong & Wang, 2023; Haque et al., 2023; Mikaeili et al., 2023; Song et al., 2023; Wang et al., 2022), as well as techniques that enable relighting and texture modification through shading cues (Gong et al., 2023; Liu et al., 2021; Rudnev et al., 2022; Srinivasan et al., 2021).

*Figure 3.* **Evolution of two physical simulations.** From left to right: (1) A rubber duck falling onto a pillow and deforming it. (2) A pirate flag waving under the influence of wind. Both simulations are performed on our own assets.

Other efforts support structural changes in the scene, such as inserting or removing objects (Kobayashi et al., 2022; Lazova et al., 2023; Weder et al., 2023), while some are tailored specifically for facial editing (Hwang et al., 2023; Jiang et al., 2022; Sun et al., 2022) or physics-based manipulation from video sequences (Hofherr et al., 2023; Qiao et al., 2022) Geometry editing within the NeRF framework has received considerable attention (Kania et al., 2022; Yuan et al., 2023; Zheng et al., 2023), highlighting the importance of explicit spatial structure for controllable scene manipulation.

Our model focuses on geometry editing and physics-driven scene manipulation. Existing editable NeRF approaches can broadly be divided into point-based, mesh-based, and physics-aware methods, each providing different trade-offs between editability, reconstruction quality, and geometric controllability.

Existing point-based methods leverage explicit geometric primitives, most notably 3D point clouds, for conditioning NeRFs. RIP-NeRF (Wang et al., 2023b) introduces a rotation-invariant point-based representation that enables fine-grained editing and cross-scene compositing by decoupling the neural field from explicit geometry. NeuralEditor (Chen et al., 2023) adopts a point cloud as the structural backbone and proposes a voxel-guided rendering scheme to facilitate precise shape deformation and scene morphing. Similarly, PAPR (Zhang et al., 2023) learns a parsimonious set of scene-representative points enriched with learned features and influence scores, enabling geometry editing and appearance manipulation. While these methods improve locality and editability compared to implicit encodings, they remain sensitive to spatial deformations because neighbourhood interpolation is typically based on Euclidean distances.

Some approaches leverage explicit mesh representations to enable NeRF editing. NeRF-Editing (Yuan et al., 2022) extracts a mesh from the scene and allows users to apply traditional mesh deformations, which are then transferred to the implicit radiance field by bending camera rays through a proxy tetrahedral mesh. Similarly, NeuMesh (Yang et al., 2022) encodes disentangled geometry and texture features at mesh vertices, enabling mesh-guided geometry editing. To reduce computational complexity, some approaches rely on simplified geometry proxies, such as coarse meshes

paired with cage-based deformation techniques (Jambon et al., 2023; Peng et al., 2022; Xu & Harada, 2022). While mesh-based methods provide intuitive geometric control and compatibility with existing graphics pipelines, they often require mesh extraction or task-specific deformation setups.

Physics-aware approaches additionally integrate physical simulations into neural rendering pipelines. VolTeMorph (Garbin et al., 2024) introduces an explicit volume deformation technique that supports realistic extrapolation and can be edited using standard software, enabling applications such as physics-based object deformation and avatar animation. PIE-NeRF (Feng et al., 2024) integrates physics-based, meshless simulations directly with NeRF representations, enabling interactive and realistic animations.

While existing approaches enable manual editing via explicit conditioning representations, they often rely on complex, task-specific pipelines (Feng et al., 2024; Jambon et al., 2023; Garbin et al., 2024). In contrast, EKS introduces a deformation-aware kernel space encoding that preserves local feature structure under affine transformations while remaining fully grid-free at inference time. Compared to prior point-based approaches (Wang et al., 2023b; Xu et al., 2022; Chen et al., 2023), our method improves interpolation stability under deformation and reduces editing artifacts while maintaining high reconstruction quality.

## 3. Preliminary

Our method, EKS, is formulated within the Neural Radiance Field framework and introduces a kernel-based latent space encoding inspired by multi-resolution hash grids and Gaussian kernel representations. In this section, we briefly review the relevant background on neural radiance fields and spatial encoding methods.

**Neural Radiance Fields** Vanilla NeRF (Mildenhall et al., 2020) represents a 3D scene as a continuous volumetric field by learning a function that maps a spatial location $\mathbf{x} = (x, y, z)$ and a viewing direction $\mathbf{d} = (\theta, \psi)$, to an emitted colour $\mathbf{c} = (r, g, b)$ and a volume density $\sigma$. Formally, the scene is approximated by a multi-layer perceptron (MLP):

$$\mathcal{F}_{\text{NeRF}}(\mathbf{x}, \mathbf{d}; \Theta) = (\mathbf{c}, \sigma), \tag{1}$$

where $\Theta$ denotes the trainable network parameters.

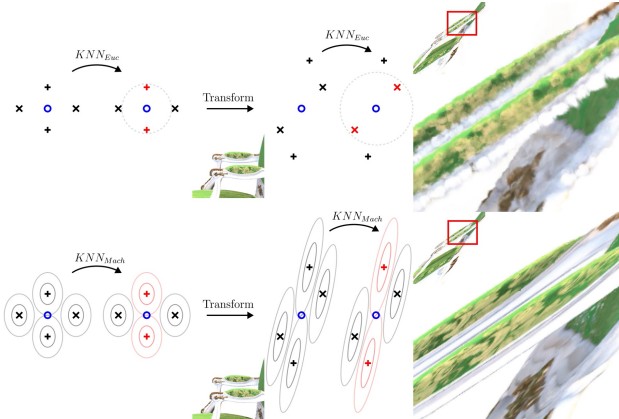

*Figure 4.* **KNN Comparisons.** Comparison of neighbourhood changes under deformation using Euclidean distance KNN (top) versus our proposed Mahalanobis distance KNN (bottom). Moving points in traditional encodings changes local neighbourhoods inconsistently, causing unstable feature interpolation. Our method preserves relative feature structure under spatial transformations and yields visibly improved results with no holes and distortions.

The model is trained using a set of posed images by casting rays from each camera pixel into the scene and accumulating colour and opacity along each ray based on volumetric rendering principles. The goal is to minimize the difference between the rendered and ground-truth images, allowing the MLP to implicitly encode both the geometry and appearance of the 3D scene.

**Hash Grid Encoding** Many NeRF variants adopt the Hash Grid Encoding (Müller et al., 2022), to improve scalability and spatial precision which captures high-frequency scene details by dividing space into multiple Levels of Detail (LoD), each with trainable parameters $\Phi$ and feature vectors $F$. These levels vary in resolution, allowing the encoding to represent both coarse and fine details. For a query point $\mathbf{x}$, the output feature vector $\mathbf{v}$ is obtained by concatenating trilinearly interpolated features from all levels, based on $\mathbf{x}$'s position within the grid

$$\mathcal{H}_{\text{enc}}(\mathbf{x}; \Phi) = \mathbf{v}(\mathbf{x}). \quad (2)$$

**Gaussian Kernels** Gaussian kernels define smooth, spatially localized basis functions in $\mathbb{R}^3$ with anisotropic support, making them well suited for continuous spatial representations and deformation-aware interpolation. Since they preserve the neighbourhood during affine transformations.

We denote a set of Gaussian kernels as

$$\{\mathcal{N}(\boldsymbol{\mu}_i, \boldsymbol{\Sigma}_i)\}_{i=1}^n. \quad (3)$$

# 4. Proposed Method

Our method, called EKS, integrates affine-equivariant transformation properties of Gaussian kernels and a neural network-based rendering procedure into a single system. Specifically, we use a set of Gaussian kernels, enhanced with a trainable latent feature vector $\mathbf{v} \in \mathbb{R}^n$. We refer to this set of Gaussians as $\mathcal{G}$.

We use a NeRF-based neural network $\mathcal{F}$ to predict colour and opacity from the nearest Gaussian features. Formally, the model is defined as:

$$\mathcal{F}(\mathbf{x}, \mathbf{d}; \mathcal{G}, \Theta) = (\mathbf{c}, \sigma), \quad (4)$$

where $\Theta$ denote the trainable network parameters. The model, alongside the standard NeRF input, takes a set of trainable Gaussians $\mathcal{G}$ and outputs colour $\mathbf{c}$ and density $\sigma$ at any query point, enabling neural rendering conditioned on nearby Gaussian features.

**Kernel Space Encoding** In point-based encodings, local neighbourhoods change inconsistently under spatial deformations, leading to unstable interpolation (see Fig. 4). Our encoding resolves this by representing latent features with anisotropic Gaussian kernels and using a Mahalanobis-distance-based interpolation that respects local geometry. Our encoding takes a set of query points $\mathbf{x}$ as input and a set of learnable Gaussians parameters $\mathcal{G}$, producing multi-resolution features. Formally, we define this encoding as:

$$\mathcal{K}_{\text{enc}}(\mathbf{x}; \mathcal{G}) = \mathbf{v}(\mathbf{x}) \quad (5)$$

Unlike the traditional *Hash Grid Encoding*, where the output depends directly on the query point $\mathbf{x}$, here the features are derived from nearby Gaussians. We select the $N$ closest Gaussians to $\mathbf{x}$ using our RT-GPS algorithm (detailed in the following section). The final feature vector is computed as a weighted interpolation of the Gaussian features using a Mahalanobis-distance-based weighting scheme:

$$\mathbf{v}(\mathcal{G}) = \sum_{i=1}^{k} w_i(\mathbf{x}, \mathcal{G}) \cdot \mathbf{v}_i, \quad (6)$$

$$w_i(\mathbf{x}, \mathcal{G}) = \exp\left(-\frac{1}{2}(\mathbf{x} - \boldsymbol{\mu}_i)^\top \boldsymbol{\Sigma}_i^{-1}(\mathbf{x} - \boldsymbol{\mu}_i)\right), \quad (7)$$

where $w_i(\mathbf{x}, \mathcal{G})$ is the interpolation weight, $k$ is the number of nearest neighbours considered, and $\boldsymbol{\Sigma}_i$ is the full anisotropic covariance of the $i$-th Gaussian kernel.

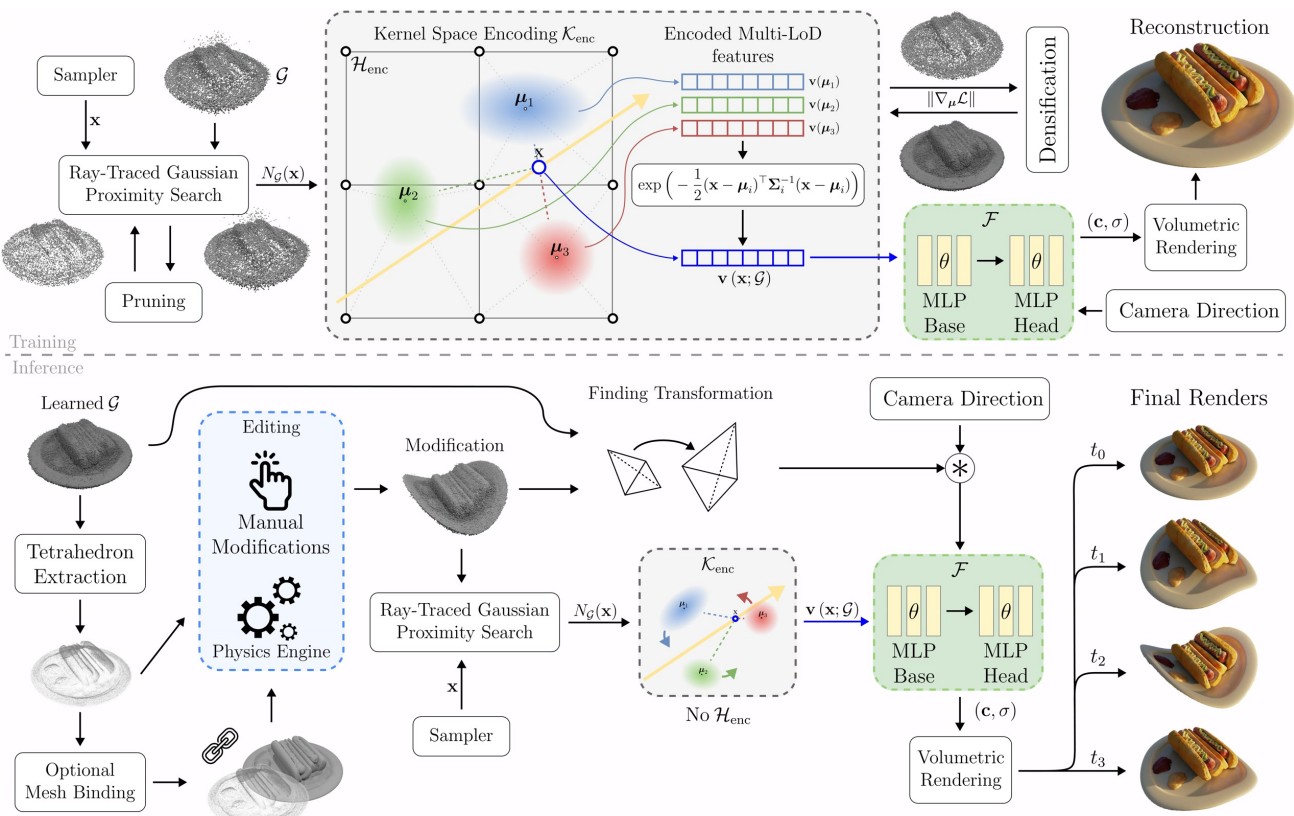

*Figure 5.* **Model overview.** Top: During training, a subset of Gaussians is selected using Ray-Traced Gaussian Proximity Search (RT-GPS), which also handles pruning. The nearest Gaussians to the sampling position $\mathbf{x}$ are passed to the Kernel Space Encoding, which interpolates their features to produce the final positional embedding $\mathbf{v}(\mathbf{x}; \mathcal{G})$. The embedding is then processed by the neural network $\mathcal{F}$ to predict colour $\mathbf{c}$ and opacity $\sigma$, which are used for volumetric rendering. Bottom: At inference time, the learned Gaussians serve as input parameters and can undergo manual or physics-driven edits. The edited Gaussians are passed through the same rendering pipeline to generate the final image, with the view-direction input to $\mathcal{F}$ adjusted by the inverse rotation of the modified Gaussians. Since the kernel space encoding is fixed after training, the auxiliary network $\mathcal{H}_{\text{enc}}$ is omitted during inference.

**Ray-Traced Gaussian Proximity Search** To achieve affine transformation equivariance, nearest-neighbour search around a query point must be performed using the Mahalanobis distance. To make this process efficient, we restrict nearest-neighbour candidates to Gaussians whose confidence ellipsoids (defined by a quantile parameter $Q$) contain the query point $\mathbf{x}$. This reduces the neighbour search to a point-in-ellipsoid test, which we approximate using circumscribed stretched icosahedra. This approach extends the RT-kNNS algorithm (Nagarajan et al., 2023). Unlike RT-kNNS, RT-GPS performs the point-in-ellipsoid test individually for each Gaussian, where the Gaussian mean corresponds to a KNN candidate for the query point $\mathbf{x}$. Following (Nagarajan et al., 2023), we trace rays originating from $\mathbf{x}$ and collect Gaussians whose confidence ellipsoids produce exactly one ray–ellipsoid intersection (see Fig. 6). A sorted hit buffer maintains up to $k$ nearest-neighbour candidates based on the squared Mahalanobis distance to $\mathbf{x}$.

**Hash Grid Feature Distillation** While hash-grid encodings are effective for representing static scenes, they do not

support precise, localized edits. Modifying vertices at lower levels of detail propagates changes to all features within the corresponding voxel, often affecting higher-resolution details and producing inconsistent, unintuitive results. To address this limitation, we introduce a Hash Grid Feature Distillation mechanism, which decouples the feature representation from the underlying grid vertices and transfers it to a set of Gaussian kernels. During training, both the hash-grid parameters $\Phi$ and the Gaussian positions $\boldsymbol{\mu}_i$ are optimized jointly, allowing the Gaussians to explore the multi-resolution feature space and shape the latent encoding. The Gaussian features $\mathbf{v}(\mathbf{x})$ are sampled from the hash-grid encoding at the kernel centres, formally described as:

$$\mathbf{v}(\mathbf{x}) = \sum_{i=1}^{k} w_i(\mathbf{x}, \mathcal{G}) \cdot \mathcal{H}_{\text{enc}}(\boldsymbol{\mu}_i; \Phi), \qquad (8)$$

At inference, we fall back to the equation 6 the hash grid is no longer needed. The Gaussians retain their learned feature vectors, which remain fixed. Since interpolation operates

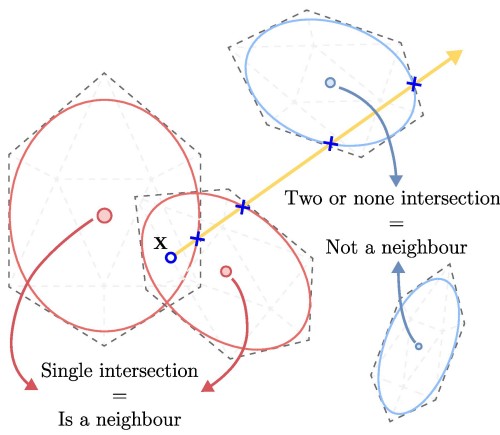

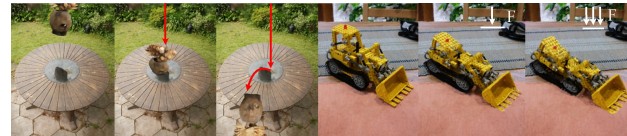

*Figure 7.* **Example edits on real-world scenes.** From left to right: (1) Physics-based simulation, showing an object falling onto a tilted table and bouncing off. (2) Physics simulation, where a force is applied to deform a plasticine dozer.

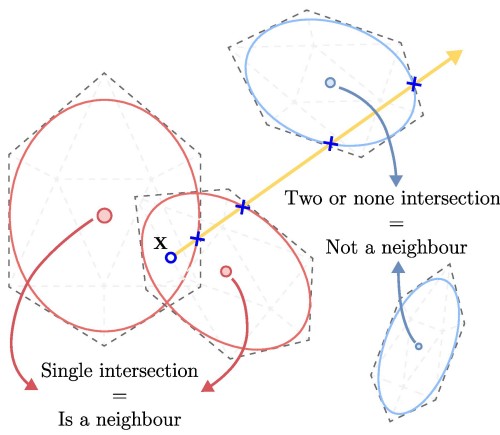

*Figure 6.* **The RT-GPS working principle.** A light ray passing through the scene is illustrated, along with its intersections with the icosahedrons. The figure highlights which Gaussians are considered neighbours and which are not.

solely over these Gaussian features, any adjustments to Gaussian positions, rotations, or scales directly modify the rendered output.

**View-Direction Restoration**   After deformation, some Gaussians may be observed from previously unseen directions. To maintain consistent appearance, we need to restore their view-dependent features as if no deformation had occurred. Chen et al. (2023) addressed this by assigning a separate local coordinate system to each point in space and tracking its transformation during deformation, which increases representation size. In contrast, our Gaussians already have principal axes that define local coordinate systems. This naturally allows us to track their spatial transformations efficiently using the Kabsch algorithm. By monitoring these axes, we can both restore view-dependent features and update the anisotropic scales of the Gaussians consistently after deformation.

**Pruning and Densification**   To enable Gaussian kernels to better represent the latent feature space, we adopt densification and pruning strategies that regulate the number of Gaussians during training. For densification, we follow the approach of (Kerbl et al., 2023), tracking Gaussian means via their gradients and cloning or splitting Gaussians accordingly. Unlike (Xu et al., 2022), we initialize the features of new Gaussians by sampling from the hash-grid encoding rather than from nearby shading information, ensuring better alignment with the latent feature field. For pruning, we track which Gaussians are actively selected as neighbours by our RT-GPS algorithm. Gaussians that are not used as neighbours for several consecutive iterations are removed, resulting in a more compact and efficient representation.

**Editing**   Thanks to the EKS feature encoding, the latent space is structured around the spatial configuration of the Gaussian kernels. This alignment allows direct edits in the coordinate space of the Gaussians, effectively translating spatial transformations into consistent latent-space manipulations. By weighting feature interpolation according to the Mahalanobis distance, we maintain affine transformation equivariance and retain the local density structure of the underlying Gaussians. As a result, the latent features remain coherent after deformation, ensuring that modifications produce smooth, stable, and physically consistent updates in the rendered scene without requiring network retraining.

In practice, for the editing task, we export Gaussians as tetrahedra, where each orthogonal arm corresponds to a principal axis of the Gaussian. This representation allows us to explicitly track how the scale and rotation of each Gaussian are affected by an edit, and additionally provides the information required for view-direction restoration. Edits can be applied directly to the tetrahedra or, alternatively, the tetrahedra can be bound to a mesh for intuitive manipulation. After editing, the modified tetrahedra are converted back into Gaussians and used as parameters for the kernel space encoding. Interpolation between these modified Gaussians then enables the system to synthesize novel views of the edited scene.

## 5. Experiments

We design our experiments to demonstrate that EKS maintains the reconstruction quality of state-of-the-art (SOTA) methods while enabling complex object modifications.

**Datasets**   Following prior work, we evaluate on the *NeRF-Synthetic* dataset (Mildenhall et al., 2020), which contains eight synthetic scenes with diverse geometry, texture, and specular properties. Additionally to synthetic data we trained our NeRF model trained on the *Mip-NeRF 360* dataset (Barron et al., 2022), comprising five outdoor and four indoor real-world 360°scenes. To further demonstrate editing capabilities, we include the *fox* scene from Instant-NGP (Müller et al., 2022), and introduce a custom set of 3D assets with deformable and articulated objects, enabling dynamic scene editing and physical interaction.

| | Chair | Drums | Lego | Mic | Materials | Ship | Hotdog | Ficus |
|---|---|---|---|---|---|---|---|---|
| Non Editable | | | | | | | | |
| INGP | 31.97 | 22.67 | 33.44 | 31.38 | 22.66 | 28.83 | 34.04 | 29.47 |
| LagHash | **35.66** | **25.68** | **35.49** | **36.71** | **29.60** | **30.88** | **37.30** | **33.83** |
| Editable | | | | | | | | |
| GaMeS | 35.73 | 26.15 | 35.57 | 35.67 | 29.89 | 30.78 | 37.58 | 34.83 |
| SuGaR | **35.83** | 26.15 | **35.78** | 35.36 | 30.00 | 30.80 | **37.72** | 34.87 |
| RIP-NeRF | 34.84 | 24.89 | 33.41 | 34.19 | 28.31 | 30.65 | 35.96 | 32.23 |
| Point-NeRF | 35.40 | 26.06 | 35.04 | 35.95 | 29.61 | 30.97 | 37.30 | **36.13** |
| Neuraleditor | 34.94 | **26.19** | 34.28 | 36.09 | **30.38** | 29.99 | 36.70 | 33.64 |
| EKS | 34.72 | 26.01 | 35.59 | **36.54** | 30.08 | **31.10** | 37.11 | 33.82 |

*Table 1.* Quantitative comparisons (PSNR) on a NeRF-Synthetic dataset showing that EKS gives comparable results with other models on static scenes.

| | Bicycle | Flowers | Garden | Stump | Treehill | Room | Counter | Kitchen | Bonsai |
|---|---|---|---|---|---|---|---|---|---|
| Static | | | | | | | | | |
| INGP | 22.17 | 20.65 | 25.07 | 23.47 | 22.37 | 29.69 | 26.69 | 29.48 | 30.69 |
| Nerfacto | 17.86 | 17.79 | 20.82 | 20.48 | 16.72 | 24.22 | 23.59 | 23.20 | 21.55 |
| Mip-NeRF | 24.37 | **21.73** | 26.98 | 26.40 | **22.87** | 31.63 | 29.55 | 32.23 | 33.46 |
| 3DGS | **25.25** | 21.52 | **27.41** | 26.55 | 22.49 | 30.63 | 28.70 | 30.32 | 31.98 |
| Editable | | | | | | | | | |
| SuGaR | 23.14 | - | 25.36 | 24.70 | - | 30.03 | 26.62 | 29.56 | 30.51 |
| GaMeS | **24.99** | 21.27 | 27.22 | 26.54 | 22.39 | 31.52 | 28.92 | 31.12 | 32.09 |
| EKS | 22.93 | 20.54 | 25.60 | 24.78 | **22.39** | 30.60 | 27.56 | 29.66 | 30.90 |

*Table 2.* Quantitative comparisons (PSNR) on a Mip-NeRF dataset.

**Baselines** We compare EKS against both static and editable point-based and Gaussian-based scene representations. For static radiance field models, we evaluate Instant-NGP (Müller et al., 2022), which introduced the hash-grid encoding and serves as the foundation of our neural field, as well as LagHash (Govindarajan et al., 2024), which augments hash-grid encodings with Gaussian primitives. While both methods achieve high reconstruction quality, they do not support scene editing.

For editable representations, we compare against RIP-NeRF (Wang et al., 2023b), Point-NeRF (Xu et al., 2022), and Neuraleditor (Chen et al., 2023), which enable scene editing using point-based NeRF formulations. We additionally include a naive plotting baseline (Chen et al., 2023) that renders a transformed dense point cloud by directly projecting points onto the camera plane using per-point opacity and view-dependent color. These baselines are selected to demonstrate that EKS not only achieves reconstruction quality comparable to or exceeding SOTA methods, while enabling editing with significantly fewer artifacts. Furthermore, we provide qualitative comparisons of physics-based simulations against PhysGaussian (Xie et al., 2024) and GASP (Borycki et al., 2024), two Gaussian-based methods designed for physical interaction, as shown in Fig. 11.

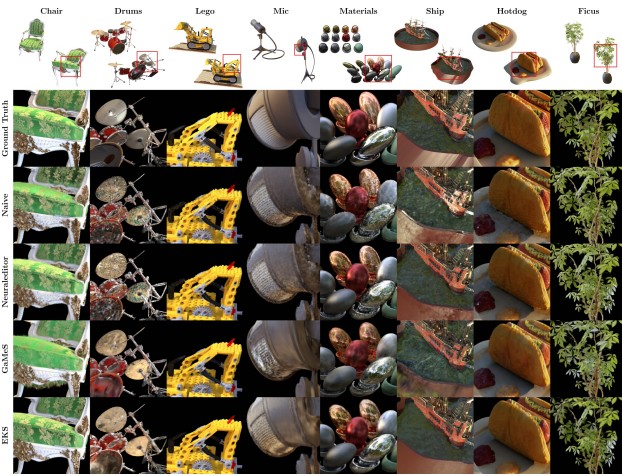

*Figure 8.* **Qualitative comparison.** Results shown on the *NeRF-Synthetic* dataset. Modified objects are in the top row. Each row compares reconstruction quality across different methods. Enlarged version ispresented in Appendix D

**Quantitative Results** We present quantitative results on *NeRF-Synthetic* and *Mip-NeRF* datasets in two settings: reconstruction of static scenes (Table 1, Table 2) and reconstruction after edits (Table 3). For static scene reconstruction, EKS achieves quality comparable to state-of-the-art editable methods, and in some cases provides the best results among methods that support editing. This demonstrates that our approach preserves rendering quality while enabling scene edits.

For edited scene reconstruction, we evaluate on the editing benchmark introduced by Chen et al. (2023), which applies handcrafted modifications to the *NeRF-Synthetic* dataset and provides ground-truth edited images. As shown in Table 3, our method consistently outperforms prior state-of-the-art approaches, including GaMeS, a purely Gaussian Splatting–based editing method.

| | Chair | Drums | Lego | Mic | Materials | Ship | Hotdog | Ficus |
|---|---|---|---|---|---|---|---|---|
| PSNR | | | | | | | | |
| Naive Plotting | 24.58 | 21.54 | 25.38 | 27.56 | 21.59 | 22.21 | 26.72 | 24.62 |
| Neuraleditor | 25.29 | 21.93 | 27.14 | 27.49 | 23.04 | 24.12 | 27.14 | 24.83 |
| GaMeS | 24.51 | 22.02 | 26.65 | 27.07 | 21.73 | 22.19 | 27.26 | 26.65 |
| EKS | **26.03** | **22.08** | **28.04** | **27.85** | **23.14** | **24.43** | **28.23** | **27.58** |
| SSIM | | | | | | | | |
| Naive Plotting | 0.930 | 0.892 | 0.904 | 0.956 | 0.867 | 0.807 | 0.930 | 0.925 |
| Neuraleditor | 0.944 | 0.900 | 0.945 | 0.958 | 0.887 | 0.832 | 0.937 | 0.927 |
| GaMeS | 0.941 | **0.914** | 0.936 | 0.960 | 0.890 | 0.811 | 0.947 | 0.947 |
| EKS | **0.957** | 0.910 | **0.961** | **0.964** | **0.911** | **0.855** | **0.962** | **0.951** |
| LPIPS | | | | | | | | |
| Naive Plotting | 0.050 | 0.107 | 0.066 | 0.053 | 0.126 | 0.187 | 0.085 | 0.072 |
| Neuraleditor | 0.041 | 0.100 | 0.038 | 0.050 | 0.103 | 0.158 | 0.078 | 0.069 |
| GaMeS | 0.039 | **0.067** | 0.035 | **0.032** | 0.077 | 0.177 | 0.046 | **0.036** |
| EKS | **0.030** | 0.071 | **0.023** | 0.036 | **0.062** | **0.143** | **0.037** | **0.036** |

*Table 3.* Quantitative comparisons (PSNR) on a (Chen et al., 2023) benchmark showing that EKS achieves best results in editing task.

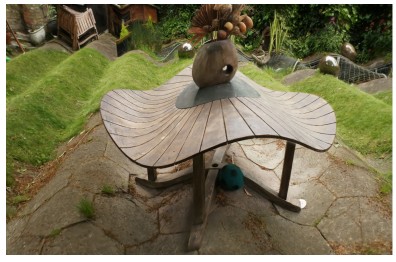 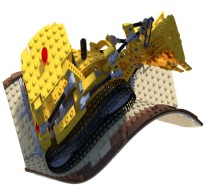

*Figure 9.* **Non-affine deformation results.** A sinusoidal transformation applied to the scene geometry to evaluate the robustness of EKS under non-affine edits. Despite violating affine assumptions, the method preserves coherent structure and maintains high visual quality.

**Qualitative Results** For qualitative evaluation, we use the editing benchmark of Chen et al. (2023) and assess the visual quality of edits across methods. We observe that EKS produces higher-quality results in the zero-shot editing setting. In particular, it better preserves fine details while yielding noticeably smoother flat surfaces. In the *Drums* scene, the gong is consistently restored without visible holes. In contrast, Neuraleditor and other point-based methods exhibit visible granularity across the image in all cases, and their edits are sometimes inconsistent, leaving holes in the reconstructed scenes. Additional artifacts are also observed, such as distortions on the plate in the *Hotdog* scene. In the Gaussian Splatting–based method GaMeS, we observe a different class of artifacts: individual Gaussians remain visible beneath the *Chair*, and in the *Ship* scene the Gaussians appear excessively scaled, causing them to bleed outside the bowl geometry. Additionally, Gaussian primitives remain visible across several scenes. In contrast, EKS avoids these artifacts and consistently produces smooth surfaces with high reconstruction quality across all evaluated scenes, as the Gaussians encode latent features rather than explicit geometry and are accessed only through local KNN-based interpolation.

**RT-GPS Evaluation** We evaluate the efficiency of Ray-Traced Gaussian Proximity Search by comparing it against both Euclidean- and Mahalanobis-distance-based nearest-neighbour search methods and Faiss (Johnson et al., 2019) library methods for GPU accelerated fast KNN search. The benchmark was conducted using 1 million Gaussian distributions, 1 million query points, and 16 nearest neighbours. As shown in Table 4, RT-GPS achieves the lowest query time among all evaluated methods while operating directly in Mahalanobis space. This efficiency stems from approximating Gaussian confidence ellipsoids with stretched icosahedra, reducing neighbour queries to efficient ray-triangle intersection tests. Compared to naive Mahalanobis-distance computation, our approach provides a substantial speedup, making it suitable for interactive editing and physics-based manipulation scenarios.

| | Fit[s] ↓ | Query[s] ↓ |
|---|---|---|
| Euclidean Distance | | |
| Naive torch Euclid | n/a | 0.2309 |
| Faiss IndexIVFFlat | 0.1183 | 0.0665 |
| Faiss IndexFlatL2 | **0.0025** | 0.1939 |
| Mahalanobis Distance | | |
| Naive torch Mahalanobis | n/a | 0.3774 |
| RT-GPS | 0.0064 | **0.0270** |

*Table 4.* Fit and query time comparison of Ray-Traced Gaussian Proximity Search

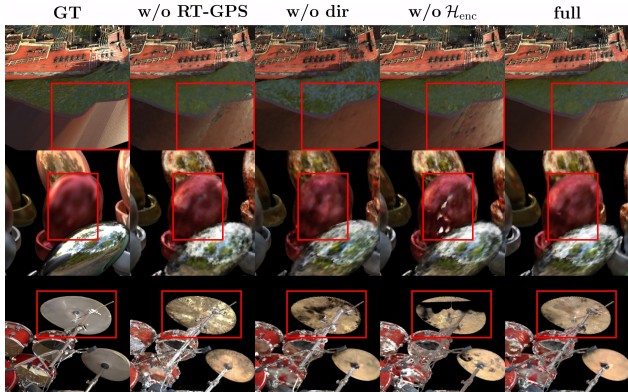

*Figure 10.* **Ablation study.** Qualitative comparison showing the effect of individual components on rendering quality.

**Physic-based Editing** We conducted a series of physics-based simulations in Blender (Community, 2018) using the mesh-driven editing mechanism described earlier. These experiments span both synthetic and real-world datasets and include diverse physical phenomena such as rigid body dynamics, soft body deformation, and cloth simulation. In these scenarios, deformations of the driving mesh were used to update the corresponding Gaussian components in real time, enabling seamless integration of physical interactions into the scene. In addition, we performed simulations following PhysGaussian (Xie et al., 2024) and compared EKS qualitatively against both PhysGaussian and GASP (Borycki et al., 2024).

The results of these simulations are illustrated in Figs 3, 2, 7, and 11. These visualizations demonstrate that EKS produces realistic and physically plausible edits across a wide range of scenarios. Whether simulating leaves falling from a plant, squashing a soft object, or draping cloth over complex geometry, our method maintains high rendering fidelity while enabling expressive and controllable scene manipulation. This highlights the potential of EKS as a flexible framework for neural scene editing driven by physical interactions.

**Ablation study** We conduct an ablation study to assess the contribution of each major component of our method. We

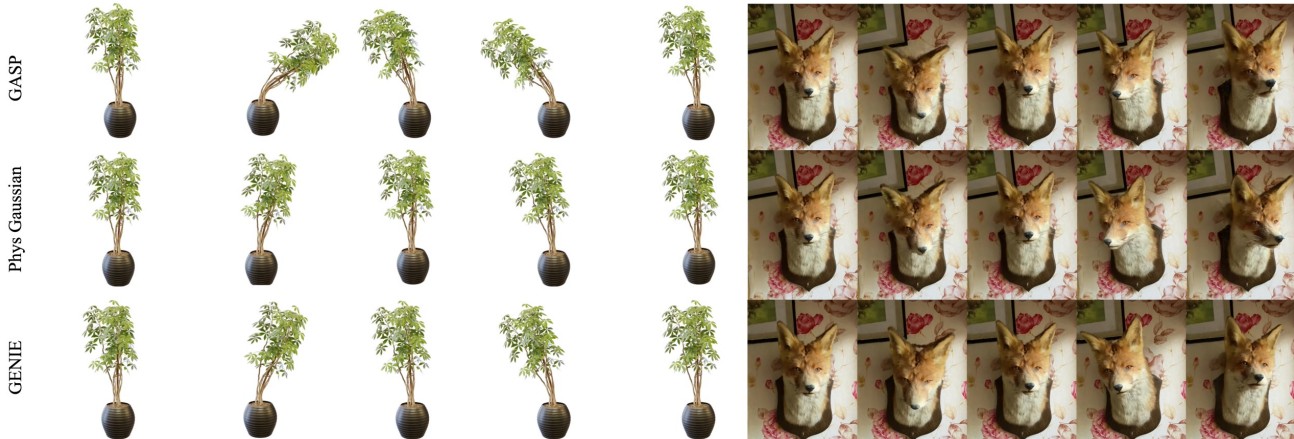

*Figure 11.* **Physics simulations with Gaussian Splatting methods.** From left to right: (1) wind simulation on the ficus plant from, (2) particle impact simulation on the fox head. Results shown for PhysGaussian (Xie et al., 2024) and GASP (Borycki et al., 2024).

evaluate variants that (1) replace RT-GPS with Euclidean KNN (w/o RT-GPS), (2) remove hash-grid feature distillation and use learned per-Gaussian features (w/o $\mathcal{H}_{enc}$), and (3) disable view-direction restoration (w/o dir). For static reconstruction, all variants achieve comparable performance, with only minor PSNR differences as shown in Table 5. In contrast, edited-scene reconstruction is more sensitive to architectural choices. Removing view-direction restoration leads to the largest performance drop, as the model fails to recover correct view-dependent appearance after deformation. Using Euclidean KNN introduces artifacts similar to point-based baselines, while removing hash-grid feature distillation has a smaller quantitative impact. However, qualitative results in Figure 10 reveal that omitting hash-grid distillation leads to visible artifacts, including holes and floating structures. Since these artifacts occur sparsely, their impact on PSNR remains limited, highlighting the importance of qualitative evaluation.

| | Chair | Drums | Lego | Mic | Materials | Ship | Hotdog | Ficus |
|---|---|---|---|---|---|---|---|---|
| **Static Reconstruction** | | | | | | | | |
| w/o RT-GPS | 34.24 | 25.83 | **36.02** | 36.13 | 29.98 | 30.72 | 36.97 | 33.21 |
| w/o $\mathcal{H}_{enc}$ | **34.72** | 25.74 | 35.74 | 35.89 | 29.89 | 30.79 | 37.09 | **33.98** |
| full | **34.72** | **26.01** | 35.59 | **36.54** | **30.08** | **31.10** | **37.11** | 33.82 |
| **Editing** | | | | | | | | |
| w/o dir | 23.80 | 21.48 | 26.07 | 27.32 | 21.15 | 21.72 | 27.70 | 25.90 |
| w/o RT-GPS | 25.58 | 21.57 | 27.55 | 27.59 | 22.91 | 24.17 | 27.28 | 26.47 |
| w/o $\mathcal{H}_{enc}$ | 25.98 | 21.83 | **28.07** | 27.70 | 23.05 | **24.51** | 28.03 | **27.64** |
| full | **26.03** | **22.08** | 28.04 | **27.85** | **23.14** | 24.43 | **28.23** | 27.58 |

*Table 5.* Ablation study of EKS reporting PSNR for static reconstruction and edited scenes.

**Non-Affine Deformations**  To further evaluate the robustness of EKS under non-affine deformations, we apply a sinusoidal transformation to the scene geometry. The re-

sulting edits are shown in Figure 9. While the deformation violates the affine assumptions of our interpolation scheme, EKS still preserves coherent local structure and produces visually plausible results with only minor artifacts.

## 6. Conclusions

We introduced EKS, an affine-equivariant kernel space encoding for Neural Radiance Fields that enables stable, localized, and deformation-aware scene editing. By representing latent features with anisotropic Gaussian kernels and aggregating them using Mahalanobis-distance-based neighbourhoods, our method preserves local feature structure under affine transformations, addressing a key limitation of point- and grid-based NeRF encodings. To retain high reconstruction quality, we proposed a training-time hash-grid feature distillation mechanism that transfers multi-resolution grid features into a compact, grid-free kernel representation at inference. This allows EKS to achieve reconstruction quality comparable to state-of-the-art NeRF models while enabling direct, intuitive editing without retraining. Across quantitative benchmarks and qualitative evaluations, our approach consistently outperforms prior editable NeRF methods, particularly after complex edits. Finally, we demonstrated that EKS naturally supports physics-driven scene manipulation, enabling realistic rigid-body, soft-body, and cloth simulations when integrated with standard physics engines. These results suggest that kernel-based, transformation-aware latent encodings provide a promising foundation for physically interactive and editable neural scene representations.

**Limitations.** While EKS performs robustly under affine transformations, extreme non-affine deformations or topology changes may still introduce artifacts. Additionally, the current method depends on RT-GPS acceleration and the quality of the Gaussian initialization, which can affect scalability and reconstruction quality in large-scale scenes.

## Acknowledgments

The work of P. Spurek was supported by the National Centre of Science (Poland) Grant No. 2023/50/E/ST6/00068. The work of M. Zieliński and D. Belter was supported by the National Science Centre, Poland, under research project no UMO-2023/51/B/ST6/01646. Some of the computations presented in this work were carried out using the infrastructure of the Poznań Supercomputing and Networking Center (PCSS).

## Impact Statement

This work contributes to the development of editable and physically interactive neural scene representations. By enabling stable, localized modifications of neural radiance fields, our method may benefit applications in robotics, simulation, virtual content creation, and scientific visualization, where controlled interaction with learned 3D environments is important. In robotics and simulation, improved scene editability can support safer testing, prototyping, and data generation in virtual settings.

The proposed method operates on 3D scene representations and does not involve human subjects, personal data, or decision-making about individuals. As such, we do not anticipate direct negative societal impacts arising from this work. Like other advances in 3D reconstruction and rendering, the technology could be repurposed for creating highly realistic synthetic content; however, this risk is shared broadly across neural rendering methods and is not unique to our approach.

Overall, we expect the primary impact of this work to be technical, enabling more controllable and physically grounded neural representations, and supporting future research at the intersection of neural rendering, simulation, and interactive 3D modelling.

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

## A. Appendix

This appendix provides additional insights and supporting material for our method. We give a formal justification of the $k$-nearest neighbour approximation used in Ray-Traced Gaussian Proximity Search, showing that distant Gaussians can be safely ignored with bounded error. We also provide extended qualitative and quantitative results to showcase the quality of our approach across various scenes.

## B. Theoretical Motivation for Ray-Traced Gaussian Proximity Search Approximation

To justify the motivation behind our *Ray-Traced Gaussian Proximity Search*, let's first recall the formula for the interpolated feature vector $\mathbf{v}(\mathbf{x})$. To begin, let's note that for the $w_i(\mathbf{x})$ appearing in the formula we have:

$$w_i(\mathbf{x}) = \begin{cases} \exp\left(-\frac{1}{2}d_M^2\left(\mathbf{x}, \mathcal{N}\left(\boldsymbol{\mu}_i, \boldsymbol{\Sigma}_i\right)\right)\right), & \text{if } i \in N \\ 0, & \text{otherwise,} \end{cases}$$

where $d_M\left(\mathbf{x}, \mathcal{N}\left(\boldsymbol{\mu}_i, \boldsymbol{\Sigma}_i\right)\right)$ is the Mahalanobis distance of the point $\mathbf{x}$ from the normal distribution $\mathcal{N}\left(\boldsymbol{\mu}_i, \boldsymbol{\Sigma}_i\right)$. Let's fix $\mathbf{x} \in \mathbb{R}^3$ and $\varepsilon > 0$. Let's consider the subset $M \subseteq N$, such that for each $i \in M$ we have:

$$d_M\left(\mathbf{x}, \mathcal{N}\left(\boldsymbol{\mu}_i, \boldsymbol{\Sigma}_i\right)\right) > \sqrt{-2\ln\left(\frac{\varepsilon}{\sum_{i \in M}|\mathbf{v}(\mathbf{x})_i|}\right)}$$

Then:

$$\left|\sum_{i \in M} w_i(\mathbf{x}) \cdot v(\mathbf{x})_i\right| =$$

$$= \left|\sum_{i \in M} e^{-\frac{1}{2}d_M^2(\mathbf{x}, \mathcal{N}(\boldsymbol{\mu}_i, \boldsymbol{\Sigma}_i))} \cdot v(\mathbf{x})_i\right| \leq$$

$$\leq \sum_{i \in M} \left|e^{-\frac{1}{2}d_M^2(\mathbf{x}, \mathcal{N}(\boldsymbol{\mu}_i, \boldsymbol{\Sigma}_i))}\right| \cdot |v(\mathbf{x})_i| =$$

$$= e^{-\frac{1}{2}d_M^2(\mathbf{x}, \mathcal{N}(\boldsymbol{\mu}_i, \boldsymbol{\Sigma}_i))} \cdot \sum_{i \in M} |\mathbf{v}(\mathbf{x})_i| <$$

$$< \frac{\varepsilon}{\sum_{i \in M}|\mathbf{v}(\mathbf{x})_i|} \cdot \sum_{i \in M} |\mathbf{v}(\mathbf{x})_i| = \varepsilon$$

Thus:

$$\left|\sum_{i \in N} w_i(\mathbf{x}) \cdot v(\mathbf{x})_i - \sum_{i \in N \setminus M} w_i(\mathbf{x}) \cdot v(\mathbf{x})_i\right| =$$

$$= \left|\sum_{i \in M} w_i(\mathbf{x}) \cdot v(\mathbf{x})_i\right| < \varepsilon$$

from which we conclude that removing the nearest neighbors from the set $M$ from the formula for $\mathbf{v}\left(\mathcal{G}_{EKS}\right)$ can alter the interpolated feature vector coordinate by no more than $\varepsilon$.

## C. Extended results

In this section, we extend the results presented in Table 1 of the main paper by additionally reporting SSIM and LPIPS metrics for both synthetic and real-world datasets.

We additionally evaluate the influence of the number of Gaussians used during initialization on the final reconstruction quality. The results, presented in Table 7, show that EKS remains relatively stable even with a substantially reduced

| | Chair | Drums | Lego | Mic | Materials | Ship | Hotdog | Ficus |
|---|---|---|---|---|---|---|---|---|
| | | | | PSNR ↑ | | | | |
| | | | | Static | | | | |
| INGP | 31.97 | 22.67 | 33.44 | 31.38 | 22.66 | 28.83 | 34.04 | 29.47 |
| LagHash | **35.66** | **25.68** | **35.49** | **36.71** | **29.60** | **30.88** | **37.30** | **33.83** |
| | | | | Editable | | | | |
| GaMeS | **35.73** | 26.15 | 35.57 | 35.67 | 29.89 | 30.78 | **37.58** | 34.83 |
| RIP-NeRF | 34.84 | 24.89 | 33.41 | 34.19 | 28.31 | 30.65 | 35.96 | 32.23 |
| Point-NeRF | 35.40 | 26.06 | 35.04 | 35.95 | 29.61 | 30.97 | 37.30 | **36.13** |
| Neuraleditor | 34.94 | **26.19** | 34.28 | 36.09 | **30.38** | 29.99 | 36.70 | 33.64 |
| EKS | 34.72 | 26.01 | **35.59** | **36.54** | 30.08 | **31.10** | 37.11 | 33.82 |

| | Chair | Drums | Lego | Mic | Materials | Ship | Hotdog | Ficus |
|---|---|---|---|---|---|---|---|---|
| | | | | SSIM ↑ | | | | |
| | | | | Static | | | | |
| INGP | 0.976 | 0.900 | 0.974 | 0.975 | 0.889 | 0.860 | 0.976 | 0.962 |
| LagHash | **0.984** | **0.934** | **0.978** | **0.991** | **0.947** | **0.892** | **0.981** | **0.981** |
| | | | | Editable | | | | |
| GaMeS | 0.987 | 0.953 | 0.982 | 0.992 | 0.952 | 0.904 | 0.985 | 0.986 |
| RIP-NeRF | 0.980 | 0.929 | 0.977 | 0.962 | 0.943 | 0.916 | 0.963 | 0.979 |
| Point-NeRF | **0.991** | **0.954** | **0.988** | **0.994** | **0.971** | **0.942** | **0.991** | **0.993** |
| Neuraleditor | 0.980 | 0.928 | 0.974 | 0.985 | 0.960 | 0.876 | 0.969 | 0.970 |
| EKS | 0.983 | 0.939 | 0.978 | 0.990 | 0.951 | 0.898 | 0.981 | 0.977 |

| | Chair | Drums | Lego | Mic | Materials | Ship | Hotdog | Ficus |
|---|---|---|---|---|---|---|---|---|
| | | | | LPIPS ↓ | | | | |
| | | | | Static | | | | |
| INGP | **0.017** | 0.094 | **0.013** | 0.027 | 0.100 | **0.119** | **0.021** | **0.030** |
| LagHash | 0.024 | **0.083** | 0.027 | **0.015** | **0.070** | 0.139 | 0.036 | 0.049 |
| | | | | Editable | | | | |
| GaMeS | **0.009** | **0.038** | 0.014 | **0.005** | 0.042 | 0.090 | 0.017 | 0.012 |
| RIP-NeRF | - | - | - | - | - | - | - | - |
| Point-NeRF | 0.010 | 0.055 | **0.011** | 0.007 | 0.041 | **0.070** | 0.016 | **0.009** |
| Neuraleditor | 0.019 | 0.061 | 0.019 | 0.016 | **0.031** | 0.075 | 0.033 | 0.033 |
| EKS | 0.011 | 0.052 | **0.011** | 0.008 | 0.036 | 0.095 | **0.016** | 0.015 |

*Table 6.* Quantitative comparisons (PSNR, SSIM, LPIPS) on a NeRF-Synthetic dataset showing that EKS gives comparable results with other models.

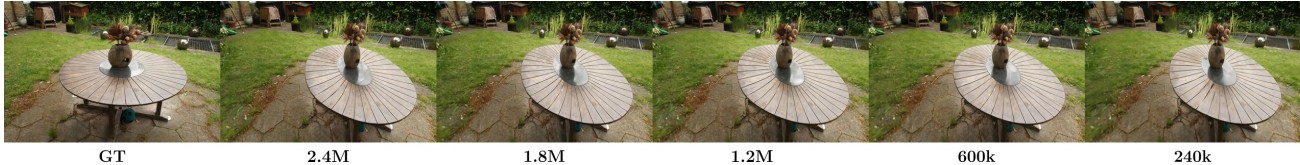

| GT | 2.4M | 1.8M | 1.2M | 600k | 240k |

*Figure 12.* **Influence of the number of Gaussians on editing quality.** Qualitative comparison of edited scenes reconstructed using different numbers of Gaussian kernels during initialization. Reducing the number of Gaussians primarily affects fine details and local surface smoothness while preserving the overall structure of the edited scene.

number of Gaussian kernels, demonstrating robustness to initialization density. Furthermore, we analyze how the number of Gaussians affects editing quality. The qualitative results shown in Figure 12 indicate that reducing the number of Gaussian kernels mainly impacts fine geometric details and local smoothness, while the overall structure of the edits remains consistent.

We also report practical runtime characteristics of EKS in Table 8. The results highlight the computational cost of editable rendering compared to the static reconstruction setting.

We additionally compare EKS against recent Gaussian reduction and compression methods (Trim the Fat (Ali et al., 2024), LP-3DGS (Zhang et al., 2024) and PUP 3D-GS (Hanson et al., 2025)) by evaluating the impact of reducing the number of Gaussian primitives to approximately 550k. The evaluation was conducted on the *Garden* scene from the *Mip-NeRF 360* dataset. Results are shown in Table 9.

| # Gaussians | PSNR ↑ |
|---|---|
| 2.4M | 25.60 |
| 1.8M | 25.48 |
| 1.2M | 25.30 |
| 600k | 24.94 |
| 240k | 24.25 |

*Table 7.* Influence of the number of initialized Gaussians on reconstruction quality for the *Garden* scene from the *Mip-NeRF 360* dataset.

| Mode | FPS ↑ |
|---|---|
| Static | 0.18 |
| Editing | 0.04 |

*Table 8.* Runtime characteristics of EKS measured in frames per second (FPS) on the *NeRF-Synthetic Lego* scene.

| Method | Full | Reduced |
|---|---|---|
| Trim the Fat | 27.81 | 26.22 |
| LP-3DGS | 27.81 | 26.46 |
| PUP 3D-GS | 27.81 | 26.21 |
| EKS | 25.60 | 24.94 |

*Table 9.* Comparison of reconstruction robustness under Gaussian reduction on the *Garden* scene from the *Mip-NeRF 360* dataset. "Reduced" corresponds to approximately 550k Gaussian primitives.

# D. High resolution qualitative results

In this section we present enlarged qualitative comparison images of our method.

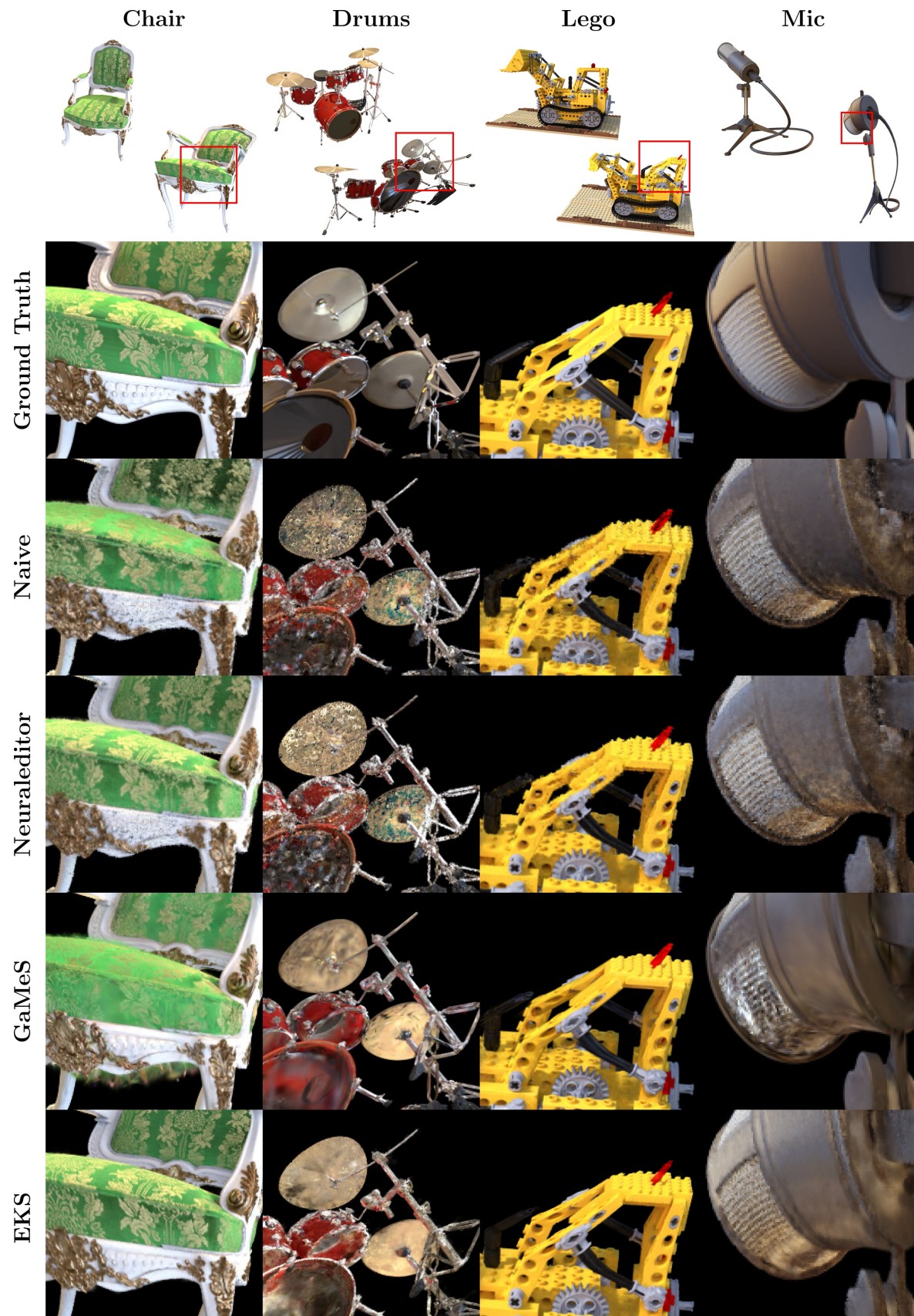

*Figure 13.* **Qualitative comparison.** Modified objects are in the top row. Each row compares reconstruction quality across different methods.

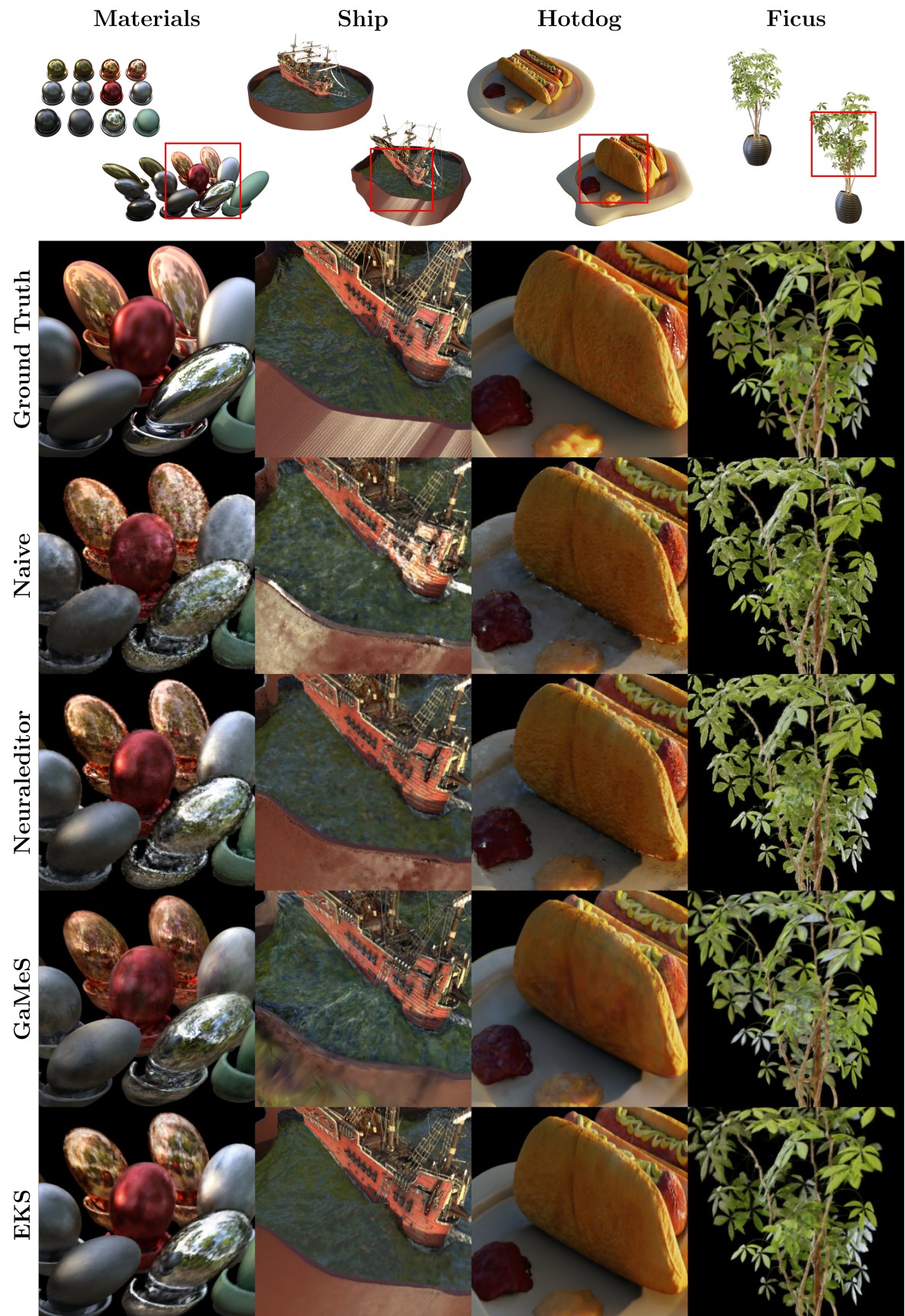

*Figure 14.* **Qualitative comparison.** Modified objects are in the top row. Each row compares reconstruction quality across different methods.

