# OpenReview forum: "Affine-Equivariant Kernel Space Encoding for NeRF Editing"
_ICML.cc/2026/Conference — ICML 2026 regular_

### Official Review · Reviewer_UNsM · 2026-03-06

**Soundness:** 2
**Presentation:** 2
**Significance:** 3
**Originality:** 3
**Overall Recommendation:** 4
**Confidence:** 4

**Summary:**

This paper proposes Affine-Equivariant Kernel Space Encoding (EKS), a spatial encoding method designed to improve the editability of Neural Radiance Fields (NeRF). The core idea is to replace traditional positional encodings or hash grid encodings with a set of anisotropic Gaussian kernels, each carrying a learnable feature vector, where features at query points are obtained through Mahalanobis-distance-weighted interpolation. To preserve the expressiveness of multi-resolution hash grids, the authors propose a training-time feature distillation mechanism (Hash Grid Feature Distillation) that transfers features from the hash grid into the Gaussian kernel field during training, allowing the representation to be fully decoupled from grid structures at inference time. Additionally, the authors design Ray-Traced Gaussian Proximity Search (RT-GPS) for efficient Mahalanobis-distance-based nearest-neighbor search, and leverage the principal axes of the Gaussian kernels for View-Direction Restoration. Experiments are conducted on the NeRF-Synthetic and Mip-NeRF 360 datasets for both static reconstruction and post-edit reconstruction evaluation, along with demonstrations of physics-simulation-driven scene editing.

**Compliance With Llm Reviewing Policy:**

Affirmed.

**Final Justification:**

The authors have addressed my concerns, and I will raise my score to 4.

**Key Questions For Authors:**

- Why are the quantitative results on the Mip-NeRF 360 dataset not presented? If performance was unsatisfactory, please report it honestly and analyze the reasons. If it was due to space constraints, this should be prioritized as supplementary content. The response to this question will affect my assessment of the method's generalization capability.
- What are the training time, inference speed, and memory consumption of EKS? Please provide quantitative comparisons against Instant-NGP, Neuraleditor, and 3DGS methods. In particular, what is the additional computational overhead of RT-GPS compared to Euclidean KNN? This will determine the assessment of this method's practicality in interactive editing scenarios.
- Why are there no quantitative comparisons against 3D Gaussian Splatting editing methods (e.g., GaussianEditor)? Is comparing only against GaMeS on the editing benchmark sufficient? Given that 3DGS inherently possesses local editability and real-time rendering capability, the lack of comprehensive comparisons against this family of methods is a significant shortcoming.
- In the ablation study, the editing PSNR of the "w/o $\mathcal{H}_{enc}$​" variant is nearly identical to the full model — how can the actual contribution of hash grid distillation be quantified?
 Is it possible to design evaluation metrics or test scenarios that more clearly highlight the differences?

**Limitations:**

The authors discuss potential societal impact (the risk of misuse of synthetic content) in the Impact Statement, which is reasonable. However, the paper's discussion of technical limitations is virtually nonexistent: there is no discussion of the method's scalability to large-scale scenes, sensitivity to the number of Gaussian kernels, RT-GPS's dependence on hardware (RT cores), or failure modes under topological changes or extreme deformations. It is recommended to add a dedicated Limitations paragraph in the main text.

**Strengths And Weaknesses:**

Strengths:
- The problem motivation of this paper is clear and meaningful. The globally entangled nature of existing NeRF encodings (positional encoding, hash grids) is indeed the core bottleneck limiting local editability, while point-cloud-based methods suffer from interpolation instability under deformations. Introducing anisotropic Gaussian kernels into the encoding space and leveraging Mahalanobis distance to achieve equivariance under affine transformations is a conceptually natural and appealing design.
- The method demonstrates good systematicity. From Gaussian kernel encoding, RT-GPS nearest-neighbor search, hash grid feature distillation, to view-direction restoration and pruning/densification strategies, it forms a fairly complete technical pipeline. In particular, the design of exporting Gaussian kernels as tetrahedra for interfacing with mesh editing tools and physics engines has practical engineering value.
- The quantitative results on the editing task (Table 2) show consistent improvements over Neuraleditor and GaMeS, and the qualitative comparisons (Figures 8, 11, 12) also demonstrate that EKS produces noticeably fewer holes and granular artifacts in post-edit reconstruction compared to baseline methods. The physics simulation demonstrations (rigid body, soft body, cloth) are diverse and showcase the generality of the method.

Weaknesses:
- The experimental evaluation is insufficient in several critical aspects. First, in static reconstruction (Table 1), EKS's PSNR is lower than Point-NeRF and even Neuraleditor on multiple scenes (e.g., Chair: 34.72 vs 35.40; Ficus: 33.82 vs 36.13), yet the authors simply gloss over this with the word "comparable" without analyzing the reasons for these gaps. Second, the editing experiments use only the single benchmark proposed by Chen et al. (2023), whose edit types are primarily rigid or simple deformations, lacking quantitative evaluation on more challenging editing operations such as large-scale non-rigid deformations or topological changes. Furthermore, the comparison with 3D Gaussian Splatting (3DGS) methods is extremely incomplete — the authors only provide qualitative comparisons with PhysGaussian and GASP in the physics simulation section (Figure 9), without quantitative metrics. Given that 3DGS has become the dominant paradigm in this field, a method that claims to simultaneously achieve high-fidelity rendering and strong editing capabilities cannot avoid comprehensive quantitative comparisons against 3DGS and its editing variants on PSNR/SSIM/LPIPS.
- The discussion of computational efficiency is entirely absent. RT-GPS involves ray tracing and ellipsoid intersection tests, whose computational overhead compared to simple Euclidean distance KNN must be significantly higher. The paper reports no data on training time, inference speed, or memory consumption, nor does it compare efficiency against baseline methods. For a method that emphasizes "interactive editing," this is an unacceptable omission. Readers cannot judge whether the method is usable in practical interactive scenarios.
- The results on the Mip-NeRF 360 dataset are not mentioned at all in the main text. The authors claim on page 6 that they trained their model on this dataset, but neither the main text nor the appendix provides any quantitative results. This omission significantly undermines the credibility of the method's generalization capability on real-world scenes.
- There are some issues with the writing. The Related Works section (Section 2) is essentially a literature enumeration, lacking systematic comparative analysis of the technical approaches between this work and existing methods. There is a spelling error in Section 4, line 200: "affine-equvariant" (should be "equivariant").

---

> ### Author Rebuttal · Authors · 2026-03-30
>
> Dear Reviewer,
>
> We sincerely thank you for your careful and constructive review. Your comments are valuable and will help us refine and strengthen our work. Our responses to your concerns and questions are provided below:
>
> **W1**. We agree that Table 1 currently lacks sufficient discussion, and we will add a more detailed analysis explaining why certain methods perform better in specific cases. Comparing only on the benchmark of Chen et al. (2023) is not ideal. We intended to also include results on the PhysGaussian benchmark, however it is not publicly available. To address this, we provide additional real-world, large-scale editing examples https://anonymous.4open.science/r/ICML-Rebuttal-16B6/README.md. We agree that including quantitative metrics for physics-based simulations would be beneficial. However, each method operates on a different number and type of primitives, which leads to different behaviors of the MPM solver even under identical initial conditions.
>
> **W2**. It is true that ellipsoid intersection tests are computationally expensive, which is why we avoid them in our approach. Instead, we approximate Gaussians using stretched icosahedrons, reducing the problem to efficient ray–triangle intersection tests. To demonstrate the performance of RT-GPS, we conducted an evaluation on 1 million Gaussian distributions with 1 million query points and 16 nearest neighbours. The results show clear advantages over both naive implementations and Faiss-based methods:
>
> ||Fit[s]$\downarrow$|Query[s]$\downarrow$|
> |-|-|-|
> ||EuclideanDistance
> |Naive Torch|n/a|0.2309|
> |Faiss IndexIVFFlat|0.1183|0.0665|
> |Faiss IndexFlatL2|0.0025|0.1939|
> ||Mahalanobis Distance
> |Naive Torch|n/a|0.3774|
> |RT-GPS|0.0064|0.0270|
>
> As shown, RT-GPS achieves the lowest query times while maintaining competitive fitting performance. This confirms that the method’s efficiency arises from both the restricted Mahalanobis-based neighbour search and the optimized ray-triangle intersection for approximated Gaussian geometry.
>
> **W3**. We acknowledge that the Mip-NeRF 360 results were missing. We will include the corresponding quantitative evaluation to better support the generalization capability of our method on real-world scenes. The results are provided in the table below:
>
> ||bicycle|flowers|garden|stump|treehill|room|counter|kitchen|bonsai|
> |-|-|-|-|-|-|-|-|-|-|
> |||||Static
> |INGP|22.17|20.65|25.07|23.47|22.37|29.69|26.69|29.48|30.69|
> |Nerfacto|23.58|21.16|25.89|25.81|22.85|30.61|27.09|29.92|30.59|
> |Mip-NeRF|24.37|21.73|26.98|26.40|22.87|31.63|29.55|32.23|33.46|
> |3DGS|25.25|21.52|27.41|26.55|22.49|30.63|28.70|30.32|31.98|
> |||||Editable
> |SuGaR|23.14|-|25.36|24.70|-|30.03|26.62|29.56|30.51|
> |GaMeS|24.99|21.27|27.22|26.54|22.39|31.52|28.92|31.12|32.09|
> |EKS|22.93|20.54|25.60|24.78|22.39|30.60|27.56|29.66|30.90|
>
>
> **W4**. We will revise Section 2 to include a systematic comparative analysis of technical approaches, emphasizing differences in physics integration, meshing requirements, and applicability to static and dynamic scenes. We will also remove the typo.
>
> **Q1**. Response in W3.
>
> **Q2**. Response partially in W2. For table with computational analysis please look Reviewer **ktpf**, W2.
>
> **Q3**. We agree that the lack of quantitative comparisons against 3D Gaussian Splatting editing methods should be addressed. Our work focuses on geometric scene manipulation, while methods such as GaussianEditor target appearance-level edits (e.g., color, texture, generative insertion), making direct comparison less straightforward. Nevertheless, we agree that comparisons within similar categories are important; therefore, we include additional evaluation against SuGaR, which addresses related geometric editing tasks.
>
> ||Chair|Drums|Lego|Mic|Materials|Ship|Hotdog|Ficus|
> |-|-|-|-|-|-|-|-|-|
> |||||Static
> |INGP|31.97|22.67|33.44|31.38|22.66|28.83|34.04|29.47|
> |LagHash|35.66.|25.68|35.49|36.71|29.60|30.88|37.30|33.83|
> |GaMeS|35.73|26.15|35.57|35.67|29.89|30.78|37.58|34.83|
> |||||Editable
> |SuGaR|35.83|26.15|35.78|35.36|30.00|30.80|37.72|34.87|
> |RIP-NeRF|34.84|24.89|33.41|34.19|28.31|30.65|35.96|32.23|
> |Point-NeRF|35.40|26.06|35.04|35.95|29.61|30.97|37.30|36.13|
> |Neuraleditor|34.94|26.19|34.28|36.09|30.38|29.99|36.70|33.64|
> |EKS|34.72|26.01|35.59|36.54|30.08|31.10|37.11|33.82|
>
>
> **Q4**. We agree with this comment and have already taken some countermeasures. In the ablation study, we explain the holes and floaters that appear in the rendered models. These inconsistencies are typically visible only over a short period of time, and quantitative metrics do not capture this phenomenon well. That is why we included a qualitative ablation in Figure 10. We acknowledge that the described effect may not be immediately noticeable. To address this, we will add arrows and boxes to draw the reader’s attention to the crucial parts.
>
> **L1**. Please look. Reviewer **ktpf**, L1.

---

> > ### Author Rebuttal · Reviewer_UNsM · 2026-04-03
> >
> > The authors have addressed my concerns.

---

### Official Review · Reviewer_ktpf · 2026-03-10

**Soundness:** 3
**Presentation:** 3
**Significance:** 3
**Originality:** 3
**Overall Recommendation:** 4
**Confidence:** 4

**Summary:**

This paper addresses the problem of editable neural radiance fields, where existing NeRF representations achieve high-quality rendering but make localized scene editing difficult due to globally entangled latent features. To solve this issue, the authors propose Affine-Equivariant Kernel Space Encoding (EKS), a  spatial encoding that represents scene features using anisotropic Gaussian kernels instead of discrete points or grid vertices. By performing feature interpolation with a Mahalanobis-distance–based weighting scheme, the method preserves local feature structure under spatial transformations, enabling stable and localized edits without retraining the NeRF model. Additionally, the authors introduce a hash-grid feature distillation mechanism that transfers multi-resolution hash-grid features into the kernel representation during training, producing a compact and grid-free model at inference time. Experiments on NeRF-Synthetic and real-world datasets show that the proposed method maintains reconstruction quality comparable to state-of-the-art NeRF models while improving editing quality.

**Compliance With Llm Reviewing Policy:**

Affirmed.

**Final Justification:**

The rebuttal has addressed my concerns and I maintain my rating.

**Key Questions For Authors:**

The paper introduces pruning and densification strategies to regulate the number of Gaussian kernels during training. However, it is unclear how sensitive the method is to the number and spatial distribution of kernels. Could the authors provide additional analysis on how kernel density affects reconstruction quality and editing stability?

The paper demonstrates several editing and physics-based simulations. Are there cases where the method fails or produces artifacts, particularly under large-scale deformations or highly complex geometry?

**Limitations:**

If the issues raised in the ’Key Questions for the Authors‘ are valid, they could potentially represent limitations that should be discussed in the paper.

**Strengths And Weaknesses:**

Strengths:

1. This design of EKS provides a transformation-aware feature representation and addresses a fundamental limitation of existing NeRF encodings, where features are globally entangled and difficult to edit locally. The method uses Gaussian kernels to model the continuous feature space, which is a straightforward yet effective idea.

2. By using Mahalanobis-distance–based interpolation over Gaussian kernels, the method preserves local feature structure during deformation. This significantly improves stability compared to point-based encodings, where neighborhood relationships change abruptly during editing and often lead to artifacts.


Weaknesses:

1. The experimental evaluation mainly compares the proposed method with point-based and Gaussian-based editable NeRF approaches, but it does not include comparisons with mesh-based NeRF editing methods (e.g., mesh-guided deformation approaches). Since mesh-based methods represent another important line of work for controllable editing, including such comparisons would provide a more comprehensive evaluation of the proposed method.

2. The paper does not provide a detailed analysis of the computational efficiency of the proposed method. The paper does not report runtime statistics such as rendering speed, inference time, or memory consumption. Since the proposed representation involves kernel interpolation and neighbor search over Gaussian primitives, it would be important to evaluate its computational overhead and scalability, especially in comparison with existing NeRF editing approaches (e.g., point-based or hash-grid methods). Including such efficiency analysis would strengthen the practical applicability of the method.

---

> ### Author Rebuttal · Authors · 2026-03-30
>
> Dear Reviewer,
>
> We appreciate your detailed review and insightful comments. Your feedback is important to us and will guide us in enhancing the paper. Below, we respond to the points you have raised:
>
> **W1**. We acknowledge that mesh-guided editing methods represent another important line of work. While our current evaluation focuses on point- and Gaussian-based editable NeRFs, we agree that including mesh-based approaches provides a more complete picture. Therefore, we include an additional comparison with NeuMesh:
>
> |Model|PSNR $\uparrow$|SSIM $\uparrow$|LPIPS $\downarrow$|
> |-|-|-|-|
> |NeuMesh|30.94|0.951|0.043|
> |EKS|35.99|0.983|0.0115|
>
> In addition to the quantitative comparison, our method differs in that it does not require an underlying mesh for initialization or editing. This simplifies the setup and enables more direct and user-friendly manipulation (e.g., using standard tools such as Blender). At the same time, our Gaussian-based representation captures geometry explicitly, making it compatible with mesh-based pipelines if needed.
>
>
> **W2**. We thank the reviewer for highlighting the importance of computational efficiency. Our RT-GPS method provides a significant improvement over existing neighbour search approaches in query time, which is critical for interactive editing and physics-based simulations. This efficiency stems from the combination of ray tracing and efficient triangle intersection operations enabled by our icosahedron-based representation. Operating in Mahalanobis space further improves the accuracy and speed of neighbour queries.
> We evaluated the performance on 1 million Gaussian distributions with 1 million query points and 16 nearest Neighbors, demonstrating clear advantages over both naive and Faiss-based approaches:
>
> ||Fit[s]$\downarrow$|Query[s]$\downarrow$|
> |-|-|-|
> ||EuclideanDistance
> |Naive Torch|n/a|0.2309|
> |Faiss IndexIVFFlat|0.1183|0.0665|
> |Faiss IndexFlatL2|0.0025|0.1939|
> ||Mahalanobis Distance
> |Naive Torch|n/a|0.3774|
> |RT-GPS|0.0064|0.0270|
>
> |Model|Training $\downarrow$|FPS $\uparrow$|GPU $\downarrow$|Training $\downarrow$|FPS $\uparrow$|GPU $\downarrow$|
> |-|-|-|-|-|-|-|
> ||||Static
> |INGP|4min 30s|1.89|3.80GB|5min 20s|0.47|3.99GB|
> |Nerfacto|5m 26s|1.91|5.13GB|5m 29s|0.72|5.86GB|
> |3DGS|6m 43s|425|4.43GB|28min 13s|121|8.30GB|
> ||||Editable
> |GaMeS|7min 8s|356|3.52GB|29min 42s|110|10.00GB|
> |NeuralEditor|103h 45min|0.08|35.61GB|n/a|n/a|n/a|
> |EKS|26h 2min|0.18|20.71GB|2h 26min|0.08|18.90GB|
>
> **Q1**. To evaluate it, we provide the following experiment on the garden scene with different numbers of Gaussians at initialization.
> |\#Gaussians|PSNR $\uparrow$|
> |-|-|
> |2.4M|25.60|
> |1.8M|25.48|
> |1.2M|25.30|
> |600k|24.94|
> |240k|24.25|
>
> Please find the link with anonymous repository showcasing editing: https://anonymous.4open.science/r/ICML-Rebuttal-16B6/README.md
>
> **Q2.** Our method performs very well under affine transformations, as the property of preserving neighbourhoods ensures that the NeRF MLP predictions remain consistent for the given points. For non-affine transformations, perfect results cannot be guaranteed. To adress this we provide somoe more examples in the link shared in Q1.
>
> **L1**. We will add a dedicated limitations section in the main paper. In particular, we will discuss the dependency on RT cores, scalability to large-scale scenes, sensitivity to the number of Gaussian kernels, and potential artifacts under non-affine transformations.

---

> > ### Author Rebuttal · Reviewer_ktpf · 2026-04-03
> >
> > Thank the authors for the rebuttal. My concerns have been addressed.

---

### Official Review · Reviewer_Bqw2 · 2026-03-11

**Soundness:** 3
**Presentation:** 4
**Significance:** 3
**Originality:** 3
**Overall Recommendation:** 5
**Confidence:** 3

**Summary:**

EKS attempts to solve the NeRF editing problem by using: Gaussians to store features that can be ray marched + Gaussians that can be converted back and forth to a tetrahedron to leverage existing physics engines to speed up the deformation process. Various techniques including RT-GPS (for finding the nearest gaussians), hash grid distillation and pruning are used to speed up the overall process of the method at multiple stages.

Overall, I find the work satisfying to read, and contains a lot of interesting techniques that could individually be inspiring or serving useful roles in even other frameworks.

**Compliance With Llm Reviewing Policy:**

Affirmed.

**Final Justification:**

The rebuttal has addressed my concerns and I maintain my rating.

**Key Questions For Authors:**

Question
1. How does the work’s performance compare with existing works like Trimming the Fat, PUP 3D-GS and LP-3DGS.
2. Are there particular benefit of this current work compared to existing works, besides the fact it relies on a different method that existing works did not exactly apply.

**Limitations:**

Societal impact is discussed (no substantial impact). The limitation of the method as is have been discussed in various parts of the paper.

**Strengths And Weaknesses:**

Strength:
1. The (multiple) ideas and techniques introduced in this paper appear novel.
2. The conversion between the Gaussians and the Tetrahedron is likely the most innovative part of the pipeline. And the information equivalence, and the ability of existing engines can use Tetrahedron efficiently is a very important feature.
3. Different acceleration techniques, whether hash grid from the Neural rendering world or the icosahedron from the graphics world are being leveraged properly at the right places in this work (a strength, not a weakness).
4. The technique introduced is consistent with the result presented, i.e. the result shows strength where it should, and the artifacts do appear wherever the afine transformation is unable to handle.

Weakness:
1. While technically very interesting, the results among various comparisons are win or loss compared to existing work.
2. The affine transformation is at the end of the day, still a linear process. While it is useful on its own, it defines a small set of deformations that could be executed.

---

> ### Author Rebuttal · Authors · 2026-03-30
>
> Dear Reviewer,
>
> Thank you for your thorough evaluation and helpful suggestions. We appreciate your feedback and will use it to improve the quality of the manuscript. Please find our responses to your questions and concerns below.
>
> **W1**. We agree with the observation. Our primary focus in this work is on editable NeRF representations, rather than on maximizing static reconstruction performance. As shown in Table 2, our approach achieves the best results in editing quality, which is the main contribution of this work.
>
> **W2**. We agree that affine transformations impose some limitations on the class of deformations that can be represented. However, our approach still reduces potential negative effects when the transformation is not affine because Gaussians encode geometry more faithfully than standalone points, preserving local neighbourhoods better and improving editing robustness.
>
> **Q1**. We have run the requested methods and provide a comparison analyzing the impact of reducing the number of Gaussians from the full Gaussian Splatting representation to a reduced setting (~550k Gaussians). The evaluation was conducted on the Garden scene from the Mip-NeRF 360 dataset.
>
> |Method|Initial|~90% Reduced|Drop|
> |-|-|-|-|
> |TrimFat|27.81|26.22|1.59|
> |LP-3DGS|27.81|26.46|1.35|
> |PUP3D-GS|27.81|26.21|1.60|
> |EKS|25.60|24.94|0.66|
>
> **Q2**. Our approach provides several key advantages. First, it enables easy initialization, as no mesh extraction or point cloud sampling is required. Second, it supports user-friendly editing, allowing seamless integration with existing tools and workflows (e.g., Blender) as well as standard Gaussian manipulation techniques. Finally, since Gaussian primitives approximate the underlying geometry, our method naturally allows future extensions where ray tracing can bypass volumetric rendering entirely, enabling significantly faster rendering. Something not feasible with point-based representations.

---

> > ### Author Rebuttal · Reviewer_Bqw2 · 2026-03-31
> >
> > The author has provided sufficient explanation or recognition of the weaknesses. The questions have been answered, with experimental result support. I therefore maintain my accept rating.

---

### Official Review · Reviewer_49WH · 2026-03-13

**Soundness:** 3
**Presentation:** 3
**Significance:** 3
**Originality:** 3
**Overall Recommendation:** 5
**Confidence:** 4

**Summary:**

This paper proposes a NeRF framework + Gaussian-based encoding to improve the editability of neural radiance fields. The key idea is to replace conventional grid- or point-based positional encodings with an affine-equivariant kernel representation based on anisotropic Gaussian primitives. By aggregating features using a Mahalanobis-distance-based neighborhood scheme, the method aims to provide more stable and localized feature interpolation under spatial transformations. To retain the expressive power of hash-grid encodings, the authors introduce a training-time distillation process that transfers multi-resolution features into the kernel representation, enabling grid-free inference. The resulting framework allows for localized and physics-driven scene editing without retraining, while maintaining rendering quality comparable to existing editable NeRF approaches. Experimental results across synthetic and real datasets suggest improved robustness and visual fidelity after editing.

**Compliance With Llm Reviewing Policy:**

Affirmed.

**Final Justification:**

(a) Fully resolved - My concerns have been adequately addressed. If you select this option, please consider adjusting your score accordingly.

I have raised my score accordingly.

**Key Questions For Authors:**

- How does the computational complexity of RT-GPS compare to standard spatial encodings in large-scale scenes?
- How sensitive is the performance to kernel density and initialization strategies?
- Can the proposed encoding naturally extend to dynamic or temporally varying NeRF settings?
- What are the practical runtime characteristics for interactive editing scenarios?

**Limitations:**

Yes

**Strengths And Weaknesses:**

## **Strengths**
- The paper tackles a meaningful and timely problem: improving the controllability and editability of neural radiance fields.
- The representation-level perspective (i.e., framing editing as an encoding design issue) is conceptually appealing and well motivated.
- The affine-equivariant kernel formulation provides an intuitive mechanism for stabilizing feature interpolation under deformation.
- Empirical results are generally convincing, with both quantitative and qualitative evidence supporting improved editing robustness.
- The ability to integrate physics-driven edits highlights the potential practical relevance of the approach.

## **Weaknesses**
- The proposed framework includes several tightly coupled components, which makes it somewhat unclear where the main gains originate.
- The computational overhead and scalability of the kernel search procedure are not analyzed in sufficient depth.
- While the experimental results are promising, evaluation remains largely limited to controlled benchmarks; broader real-world validation would strengthen the claims.
- The contribution primarily lies in a novel combination and reinterpretation of existing ideas, which may affect perceived novelty depending on reviewer expectations.
- The relationship and distinctions between this approach and Gaussian-splatting-based editing methods could be clarified more explicitly.

---

> ### Author Rebuttal · Authors · 2026-03-30
>
> Dear Reviewer,
>
> Thank you for your thoughtful and detailed feedback. We appreciate your insights and believe they will help improve the clarity and strength of our paper. We address your comments and questions in detail below:
>
> **W1**. We appreciate this observation. To disentangle the contributions of individual components, we include an ablation study (Table 3, Fig. 9 from the main paper) that evaluates the impact of RT-GPS, hash-grid feature distillation, and view-direction restoration. The results show that while all components contribute to performance, the primary gains in editing quality stem from the proposed kernel space encoding. Accordingly, our evaluation focuses on demonstrating the effectiveness of this encoding in enabling stable and localized edits.
>
> **W2**. Our RT-GPS method significantly outperforms existing approaches in query time. This improvement is primarily due to the combination of ray tracing and the efficient triangle intersection operations enabled by our icosahedron-based representation. We evaluated the method on 1 million random Gaussian distributions, 1 million query points, and 16 nearest neighbours. Operating in Mahalanobis space is a key factor, in our method.
>
> ||Fit[s]$\downarrow$|Query[s]$\downarrow$|
> |-|-|-|
> ||EuclideanDistance
> |Naive Torch|n/a|0.2309|
> |Faiss IndexIVFFlat|0.1183|0.0665|
> |Faiss IndexFlatL2|0.0025|0.1939|
> ||Mahalanobis Distance
> |Naive Torch|n/a|0.3774|
> |RT-GPS|0.0064|0.0270|
>
> **W3**. We provide the results on a real-world static reconstruction task in the table below. Our method achieves competitive reconstruction quality compared to other baselines. This aligns with our focus: the main goal of our work is not to maximize static reconstruction scores, but to enhance editability and consistency in editable scene representations.
>
> ||bicycle|flowers|garden|stump|treehill|room|counter|kitchen|bonsai|
> |-|-|-|-|-|-|-|-|-|-|
> ||||||Static|
> |INGP|22.17|20.65|25.07|23.47|22.37|29.69|26.69|29.48|30.69|
> |Nerfacto|17.86|17.79|20.82|20.48|16.72|24.22|23.59|23.20|21.55|
> |Mip-NeRF|24.37|21.73|26.98|26.40|22.87|31.63|29.55|32.23|33.46|
> |3DGS|25.25|21.52|27.41|26.55|22.49|30.63|28.70|30.32|31.98|
> ||||||Editable|
> |EKS|22.93|20.54|25.60|24.78|22.39|30.60|27.56|29.66|30.90|
>
> **W4**. We thank the reviewer for pointing this out. While our work combines and reinterprets existing ideas, we note that this combination is carefully designed to maximize efficiency and editability. In particular, as highlighted by reviewer Bqw2, the conversion between Gaussians and Tetrahedra represents a highly innovative component, enabling information equivalence while leveraging existing engines efficiently. Furthermore, different acceleration strategies, whether hash grids from neural rendering or icosahedron-based methods from graphics, are applied appropriately, demonstrating that our approach strategically integrates existing techniques rather than merely combining them, which reinforces the novelty of our framework.
>
> **W5**. We thank the reviewer for the comment. Gaussian-splatting-based editing methods are known to be sensitive to outliers, long Gaussians, or sparse features, which can lead to visible artifacts. In contrast, our encoding, combined with the neural network regularization, reduces small artifacts and improves editing robustness and consistency. This distinction highlights how our approach addresses practical limitations of standard Gaussian-splatting methods while maintaining comparable rendering quality.
>
> **Q1**. Standard spatial encodings (e.g., positional encoding and hash grid encodings) are computationally more efficient than RT-GPS. However, they do not support editable scene representations. Within the class of editable approaches, RT-GPS demonstrates the lowest computational cost. This is proven in Table in W2.
>
> **Q2**. To evaluate it, we provide the following experiment on the garden scene with different numbers of Gaussians at initialization.
> |\#Gaussians|PSNR $\uparrow$|
> |-|-|
> |2.4M|25.60|
> |1.8M|25.48|
> |1.2M|25.30|
> |600k|24.94|
> |240k|24.25|
>
> **Q3**. Yes, the proposed encoding can naturally extend to dynamic or temporally varying NeRFs. For instance, it can be combined with methods such as D-MiSo (Waczyńska et al.), allowing our encoding techniques to handle dynamic scenes effectively. Extending our approach to dynamic NeRFs is part of ongoing work.
>
> **Q4**. We evaluated running characteristics on chair model for our method one neuraleditor edit. They are as follow:
> |Mode|FPS $\uparrow$|
> |-|-|
> |Static|0.23|
> |Editing|0.04|

---

### Decision · Program_Chairs · 2026-04-30

**Decision:**

Accept (regular)

**Comment:**

The paper proposes a novel affine-equivariant kernel space encoding to aggregate the features of local 3DGS. The authors have further shown that hash grid distillation enables efficient and compelling rendering quality at inference time. All reviewers agree that the problem is well-motivated and the affine-equivariant kernel space encoding idea is plausible. During the discussion phase, several reviewers raised questions on missing computational efficiency analysis, limited experimental scope, and insufficient discussion of static reconstruction gaps. The rebuttal seems to have adequately addressed these points with additional benchmarks and comparisons, and all reviewers confirmed their concerns were fully resolved with positive ratings (5, 5, 4, 4). In the final version, the authors promise to revise the related work section, add a dedicated limitations discussion, and improve figure annotations.